# Human and Animal Brucellosis in Nigeria: A Systemic Review and Meta-Analysis in the Last Twenty-One Years (2001–2021)

**DOI:** 10.3390/vetsci9080384

**Published:** 2022-07-26

**Authors:** Kabiru O. Akinyemi, Christopher O. Fakorede, Kehinde O. Amisu, Gamal Wareth

**Affiliations:** 1Department of Microbiology, Lagos State University, Km 15, Badagry Expressway, Lasu Post Office, Ojo, P.O. Box 0001, Lagos 102101, Nigeria; kriskorede@yahoo.co.uk (C.O.F.); kehinde.amisu@lasu.edu.ng (K.O.A.); 2Friedrich-Loeffler-Institut (FLI), Institute of Bacterial Infections and Zoonoses, Naumburger Str. 96a, 07743 Jena, Germany; gamal.wareth@fli.de

**Keywords:** human and animal brucellosis, *Brucella*, Nigeria, prevalence and seroprevalence, systemic review and meta-analysis

## Abstract

**Simple Summary:**

Brucellosis caused by *Brucella* spp. is transmitted by direct or indirect contact with infected animals or their secretions and through the consumption of infected animal meat and unpasteurized milk/milk products. Brucellosis is classified as one of the top neglected zoonosis by the World Health Organization (WHO), and despite this, it does not attract the appropriate attention it requires from both the Federal and State ministries of health in Nigeria. Currently, there is a lack of coordinated national data on the prevalence and distribution of human and animal brucellosis. Thus, published research works between 2001 and 2021 were studied based on set criteria to estimate the burden and distribution of brucellosis in Nigeria. The results of the national seroprevalence of human and animal brucellosis were 17.6% (554/3144) and 13.3% (8547/64,435), respectively. Specifically, 15.8% (7178/45,363) seroprevalence of brucellosis was recorded in northern Nigeria as against 8.7% (1902/21,740) in the southern part. *Brucella abortus*, *B. melitensis*, *B. suis*, and *B. canis* were reported in 27 of the 36 states. Improved sanitation at the abattoirs, use of personal protective equipment by animal handlers, vaccination of animals against brucellosis, and ranching of animals to curb the spread of the disease should be paramount to all stakeholders.

**Abstract:**

The global burden of human and animal brucellosis remains enormous. The disease, which is endemic in Nigeria, lacks appropriate attention and national data. This review estimated the burden and distribution of human and animal brucellosis in Nigeria in the last twenty-one years (2001–2021). Publications reporting the detection of brucellosis in Nigeria were sorted from different search engines, including PubMed, ResearchGate, Scopus, and Google Scholar, to generate data on its prevalence, spatial distribution, and predisposing factors. The results of the national seroprevalence of human and animal brucellosis as revealed in this study were 17.6% (554/3144) and 13.3% (8547/64,435), respectively. Specifically, 15.8% (7178/45,363) seroprevalence of brucellosis was recorded in northern Nigeria as against 8.7% (1902/21,740) seroprevalence in the southern part. It also indicated that 78.7% of the detected brucellae were un-typed. The *Brucella* species detected were *B. abortus* (15.2%), *B. melitensis* (4%), *B. suis* (1.8%), and *B. canis* (0.4%). This study revealed that brucellosis is endemic in Nigeria. Culture and molecular methods for detecting brucellosis and reports on antimicrobial susceptibility testing remain a conjecture. This review will help researchers redirect their research focus and serve as a guide for policymakers on measures for managing brucellosis in Nigeria.

## 1. Introduction

Brucellosis, which is also referred to as the travel-related infectious disease “undulant fever”, “Mediterranean fever”, “gastric remittent fever”, or “Malta fever”, is a zoonotic disease caused by intracellular Gram-negative coccobacilli bacteria of the genus *Brucella* [1,2,3]. The disease is distributed globally, affecting humans, a wide range of wild animals, and economically viable domestic livestock such as cattle, goats, sheep, donkeys, camels, swine, dogs, etc. [4]. Currently, 12 species of the genus *Brucella* are accepted; however, only *B. melitensis*, *B. abortus*, *B. suis*, and in rare cases, *B. canis* are characterized as human pathogens. The global burden of human brucellosis remains very large. The organism causes more than 500,000 infections per year worldwide [3]. The socio-economic impact of brucellosis is enormous and is higher in developing countries than the developed countries, with an estimated 3.5 billion individuals at risk of being infected with one or more *Brucella* spp. and a high morbidity rate in both humans and animals [5,6]. Brucellosis is highly contagious. It is transmitted by direct or indirect contact with infected animals or their secretions and through the consumption of infected animal meat and products such as unpasteurized milk/milk products [1,7]. The risk of acquiring the disease has been attributed to a certain occupation (occupational hazard) but most especially among livestock caregivers. People who work with animals and are constantly in contact with blood, placenta, foetuses, and uterine secretions have an increased risk of contracting the disease. This method of transmission primarily affects farmers, butchers, hunters, veterinarians, and laboratory personnel [4]. However, in endemic areas, human brucellosis has serious public health consequences. The organism can enter the human body through breaks in the skin, mucous membranes, conjunctiva, respiratory and gastrointestinal tracts, resulting in systemic infection with acute and chronic phases [3,8]. The disease may persist as relapse, chronic localized infection, or delayed convalescence. Symptoms of the disease include but are not limited to fever or chill, arthralgia or arthritis, sweating, hepatomegaly, splenomegaly, anorexia, asthenia, fatigue, weakness, pallor, diarrhea, jaundice, lymphadenopathy, rash, and malaise [1,8]. Urbanization and the subsequent expansion of animal industries, coupled with a lack of good hygienic practices, especially in animal husbandry and food handling, partly account for brucellosis remaining a public health problem [1]. It affects people of all races, age groups, and both sexes. Although many countries have made great progress in controlling the disease, *Brucella* infection persists in domestic animals in some regions, and consequently, transmission to the human population is imminent. Brucellosis in livestock is mostly a reproductive disease characterized by infertility, late abortion, retained foetal membranes, and impaired productivity [9]. Brucellosis infects many species, especially cattle, sheep, goats, and pigs. Different *Brucella* types infect different species preferentially. The disease is still widely distributed in Africa, especially in areas with large animal populations [10].

Brucellosis, currently classified as one of the top neglected zoonosis by the World Health Organization (WHO), does not attract the appropriate attention it requires from both Federal and State ministries of health in Nigeria. According to the United Nations Department of Economic and Social Affairs, Population Division 2019 [11], Nigeria is the most populous country in Africa and ranked 7th in the world with over 201 million people. A projection of its population is expected to rise to about 401 million people by 2050, with an estimated livestock population of 20.49 million cattle, 23.07 million sheep, 28.07 million goats, 6.54 million pigs, 18,200–90,000 camels, and 210,000 horses [12,13]. Livestock slaughtered at the different abattoirs in Nigeria for human consumption are not usually screened for brucellosis. Free-range domestic animals are also commonplace in Nigeria; pet animals, i.e., dogs, cats, and livestock, i.e., goats, sheep, and cattle, move freely amongst the people. The incidence of brucellosis in Nigeria is under-reported. Currently, there is insufficient epidemiological data on Nigeria’s prevalence and distribution of human and animal brucellosis. This review of the literature on reported cases of brucellosis in Nigeria was undertaken to determine the true prevalence and distribution of the disease across the country and to make valuable suggestions that will serve as a guide for the implementation of measures for sustainable management of this disease in Nigeria.

## 2. Materials and Methods

### 2.1. Study Area

The study was conducted in Nigeria, made up of 6 geopolitical zones (North-East, North-West, North-Central, South-East, South-West, and South-South). Each zone comprises 6 states, except for the South-East with 5 states and North-West with 7 states, respectively. Nigeria has 36 states and a federal capital territory (Abuja). The states within each geopolitical zone have similar ethnic groups, political history, culture, spoken language, and environmental conditions. The North-West covers an area of 242,425 km^2^ or 25.75% of the country’s total land mass. The North-East occupies about 272,451 km^2^. The North-Central covers an area of approximately 193,088 km^2^. Meanwhile, in southern Nigeria, the South-East occupies an area of approximately 29,525 km^2^, while the South-West covers a land mass totalling 79,665 km^2^, and the South-South occupies approximately 84,587 km^2^.

### 2.2. Search Strategy and Data Acquisition

This systematic review was performed using public scientific databases, which include ResearchGate, Google Scholar, Scopus, and PubMed, to retrieve original articles reporting prevalence, serotypes, and antibiotic resistance in brucellae isolated from humans and animals in the last twenty-one years from 1 January 2001 to 31 December 2021 from all the six geopolitical zones of Nigeria. Publications reporting full research articles and letters on detecting *Brucella* by culture, serology, and molecular methods published in English languages were accessed within the six geopolitical zones. The following search strings were also applied: “*Brucella*”, “brucellae”, “brucellosis”, “undulant fever”, “Mediterranean fever”,” gastric remittent fever” or “Malta fever”, “humans”, “animals”, febrile, “abortion”, “prevalence”, “incidence”, “sensitivity testing”, “treatment”, “antibiotic susceptibility”, and “antimicrobial activity”. The list of the 36 states of Nigeria, including the six geopolitical zones, was used as the basis for searching.

### 2.3. Selection Criteria

Online articles/abstracts (full text) were reviewed individually to include articles that performed serological methods and cultural and molecular techniques on human and animal samples to detect *Brucella* in Nigeria. Relevant articles were retrieved and thoroughly reviewed using inclusion and exclusion criteria set for this study. The reference sections of retrieved publications were also reviewed thoroughly in search of further potential articles for inclusion.

### 2.4. Inclusion Criteria

Research articles on cross-sectional and cohort studies were included if they reported the detection of *Brucella* from human and animal samples if samples were collected from a named hospital or farms within a named community, state, or geopolitical zone if the study subjects were recruited in Nigeria.

### 2.5. Exclusion Criteria

Studies were excluded if the reported brucellosis-associated diseases were not based on serology, culture, or molecular methods and if *Brucella* species were not from human and animal samples. Reports that are based on clinical diagnosis, systematic reviews, ecological correlation, books, and book chapters were excluded from the studies. Research works conducted within Nigeria and outside Nigeria whose study subjects were not recruited in Nigeria were also excluded. Duplicated articles were verified and removed using two de-duplication options, which included the Mendeley citation manager and Ovid multifile search.

### 2.6. Data Extraction

Descriptive and quantitative variables relevant to the inclusion criteria set for this systemic review were extracted from each of the selected articles. A Microsoft Excel 2013 workbook was used to create a standard template, with each column corresponding to the investigating parameters. To ensure accuracy, double data extraction and entry were performed. Information extracted from the articles included state, geopolitical zone, year of sampling/study, name of the author (first), publication date, date of sample collection, source of sample (humans and animals), types of sample, sample size, number of human participants, number of animal participants, types of animals, testing method, number of positive samples from serology, culture and molecular methods, and *Brucella* biotypes, the number of isolates, species of *Brucella* isolated and antibiotic resistance profile.

## 3. Results

### 3.1. Data Analysis

Data drawn from this study were cleaned, and a descriptive analysis was performed. Online databases were searched; these included Google Scholar, ResearchGate, Scopus, and PubMed from 1 January 2001 to 31 December 2021, which produced 3149 articles. These articles were assessed for relevance based on the inclusion criteria. In all, 150 articles were found to be eligible. A further quantitative review of the articles with the removal of duplicate studies yielded only 99 articles (Figure 1). Reported cases of brucellosis were directly assessed from the literature. Prevalence was calculated in this study as the percentage of positive samples for *Brucella* in the total samples recorded.

### 3.2. Prevalence of Human and Animal Brucellosis in Nigeria

A total of 99 research articles spanning 27 states (Plateau, Benue, Kwara, Kogi, Niger, Nasarawa, Borno, Gombe, Yobe, Bauchi, Adamawa, Jigawa, Kano, Kaduna, Katsina, Sokoto, Taraba, Zamfara, Enugu, Anambra, Ebonyi, Edo, Cross River, Akwa-Ibom, Lagos, Ogun, and Oyo states) and the federal capital territory (Abuja) from online database who reported *Brucella* infection from 1 January 2001 to 31 December 2021 were reviewed. Brucellosis cases were reported in six geo-political zones in Nigeria; in NC [14,15,16,17,18,19,20,21,22,23,24,25,26,27,28,29,30,31,32,33,34,35,36,37], NE [15,36,38,39,40,41,42,43,44,45,46,47,48,49,50,51,52,53,54,55,56,57,58,59,60,61,62], NW [15,16,36,40,48,63,64,65,66,67,68,69,70,71,72,73,74,75,76,77,78,79,80,81,82,83,84,85,86,87,88], SE [36,62,89,90,91,92,93,94], SS [62,95,96,97] and SW [15,35,62,98,99,100,101,102,103,104,105,106,107,108,109,110] (Appendix A). Of the 99 publications, 11 investigated brucellosis in humans, 85 investigated *Brucella* infection in animals, and 3 investigated *Brucella* infection in humans and animals, respectively. In this study, 554 out of the 3144 were positive for human brucellosis, representing a 17.6% national prevalence. The highest prevalence of human brucellosis was observed in the South-South with a prevalence of 30.7% from a publication, while the lowest prevalence was observed in the South-West (SW) with a prevalence of 7.9%. Others include North-Central (NC) 17.1%, North-East (NE) 13.7%, North-West 8.8%, and a prevalence of 28.6% were recorded in the South-East (Appendix A and Table 1). In all, brucellosis was investigated in nine different animal hosts, i.e., cattle, donkeys, camels, horses, goats, sheep, pigs, chickens, and dogs. The national seroprevalence of brucellosis in cattle stood at 12.2% (5187/42,508), with an odds ratio of 0.1390, Z-statistics of 120.853, and a *p*-value of 0.0001 at a 95% confidence level from 43 publications. The seroprevalence of *Brucella* infection in donkeys was 10.2% (215/2101), with an odds ratio of 0.2427, z-statistic of 16.247, and *p*-value of 0.0001 at 95% CL from four publications. The seroprevalence in camels was 20.9% (514/2459) from five publications. The seroprevalence in horses was 10.7% (106/988) from eight publications, while in goats, seroprevalence was 10.2% (846/8309) from 20 publications. The seroprevalence in sheep was 23.3% (912/3908) from 22 scientific articles, and in pigs, seroprevalence was 28.3% (276/975) from five publications. Others are chicken, including turkey, guinea fowl, and avian spp., with 8.4% seroprevalence (47/1555) from three publications, and the seroprevalence in dogs was 18.9% (444/2291) from eight publications (Table 2).

### 3.3. Regional Prevalence of Brucellosis in Common Domestic Animals in Nigeria

The prevalence of brucellosis in the different domestic animals investigated varies from region to region. In cattle, the prevalence of brucellosis in North-Central (NC) Nigeria was 8.6% (277/3213), while the North-East (NE) was 22% (1154/5253), North-West (NW) was 13.3% (2617/19,648), South-East (SE) was 3.3% (51/1567), South-South (SS) was 40.9% (151/369), and South-West (SW) was 7.5% (937/12,458). Only the NE geopolitical zones reported an incidence of brucellosis in donkeys with a prevalence of 10.2% (215/2101). Similarly, the NE and NW geopolitical zones are the only regions that reported seroprevalence of brucellosis in camels, with a prevalence of 31.4% (398/1267) and 9.7% (116/1192), respectively. Prevalence of brucellosis in horses was 27.9% (24/86) in the NC, 16% (16/100) in the NE, 13.5% (54/400) in the NW, and 3% (12/402) in the SE. There were no reports on brucellosis in horses from SS and SW Nigeria. Moreover, in goats, a prevalence of 19.2% (410/2134) was observed in the NC region, while an overall prevalence of 9% (75/831) was seen in the NE, 13.8% (303/2191) in the NW, 3.5% (12/340) in the SE and 1.6% (46/2813) in the SW. In contrast, the SS has no report. In sheep, a 24.6% (322/1308) seroprevalence was observed in the NC, 24.6% (14/57) in the NE, 22.9% (562/2449) in the NW, and 14.9% in the SW (14/94), with no record in the SE and SS. The highest occurrence of *Brucella*-associated infection within the geopolitical zones was observed in pigs from SS Nigeria, with 100% (55/55) prevalence, as multiple *Brucella* serotypes were reported per sample. Meanwhile, the NC recorded a 34% (125/369) prevalence rate, which in the SE was 0.6% (2/352) and in the SW was 0% (0/200). Seroprevalence of brucellosis in chickens was 3.6% (10/275) in the NW, 2.5% (18/730) in the NE, 3.4% (14/410) in the SE and 3.6% (5/140) in the SW. Seroprevalence of brucellosis as observed in dogs was 23.4% (115/492) in the NC, 20.3% (76/374) in the NE, 21.5% (43/200) in the NW, 27.6% (34/123) in the SE, 0% in the SS and 16% (176/1102) in the SW (Table 2).

### 3.4. Prevalence of Brucella Infection According to Sample Type

In this study, 7826 out of the 58,561 blood samples collected from different animals across the country were positive for brucellosis, giving a seroprevalence of 13.4% in animals nationwide. However, in the milk samples, 10.2% (574/5608) seroprevalence was obtained, and 12.4% (89/720) seroprevalence was observed in vaginal swabs. Moreover, a 5% (1/200) prevalence was recorded in aborted fetuses, and a prevalence of 29% (50/170) was recorded in lymph nodes. The highest occurrence of *Brucella* infection was observed in hygroma fluid, with a 50% (7/14) prevalence (Table 3).

### 3.5. Diagnostic Methods Used in the Detection of Human and Animal Brucellosis in Nigeria

The Rose Bengal rapid test (RBPT) was used by the majority of the authors as the preferred diagnostic method and was reported by 11 of the 14 publications on the detection of brucellosis from human samples, while 74/99 publications reported RBPT as the detection method in animal samples. Other diagnostic tools used across the country were the serum agglutination test (SAT), which was used in 2 publications on humans and 28 from animals, the competitive enzyme-linked immunosorbent assay (cELISA), which was used in 22 publications on animals, indirect enzyme-linked immunosorbent assay (iELISA), which was used in 2 publications on humans and 10 on animals, immunoglobulin M (IgM), which was used in 2 publications each on humans and animal, and immunoglobulin G (IgG), which was used in 3 publications on humans. On the other hand, 1 publication applied the immunocomb, 10 applied the milk ring test (MRT), 1 used the complement fixation test (CFT), 2 used the micro-titre serum agglutination test (MSAT), 1 used the Wright agglutination rest (WAT), 6 used the lateral flow assay (LFA), 2 used the micro-agglutination test (MAT), and 2 applied the rapid slide assay (RSA with variation in disease prevalence, based on the different diagnostic methods employed (Figure 2)).

### 3.6. Prevalence and Diversity of Brucella Species from Humans and Animals within the Six Geopolitical Zones of Nigeria

A total of 9101 *brucellae* were reported from **68,238** human and animal samples investigated. The study revealed that only four *Brucella* species have been documented in Nigeria. These *Brucella* spp. are *B. abortus*, *B. melitensis*, *B. suis*, and *B. canis*. This study has also demonstrated that only two *Brucella* species were studied nationwide. *B. abortus* represents 2.4% national prevalence, and *B. melitensis* represents 12.6% national prevalence and has been reported so far in human *Brucella* infection, while 85% of the brucellae were un-typed. Of the two *Brucella* spp. detected in humans, two regions (NE and NW) reported *B. abortus* while SS reported *B. melitensis* (Appendix A and Table 1). In animals, of the 5197 positive cattle samples, 4197, representing 80.8%, were un-typed *Brucella* species. *B. abortus* is the most common *Brucella* biotype reported, representing 19.1% (995) of the total *brucellae* recorded, while 29 of the *brucellae*, representing 0.6%, were observed as *B. melitensis.* Others include camel, where 508/514, representing 98.8%, were un-typed *Brucella* spp., while the remaining 1.2% were observed to be *Brucella abortus*. All the *Brucella* spp. detected in donkey (215/215) and chicken (47/47) were un-typed. The majority (79.2%) of the *Brucella* spp. detected in horses were un-typed, while 22 of the 106 positive samples representing 20.8% were *B. abortus*. Un-typed *Brucella* species is also common in goats (78.5%), sheep (70.9%), pigs (31.9%), and dogs (60.8%). The highest occurrence of *B. abortus* was observed in dogs, with a prevalence of 31.3%. Among goats, the occurrence of *B. abortus* was 10% (84), *B. melitensis* was 8.1% (68), and *B. suis* was 3.4%. In sheep, the occurrence of *B. abortus* was 9% (82), *B. melitensis* was 15.7% (143), and *B. suis* was 4.4% (40). The detection of *B. abortus* in pigs was 15.6% (43), while the detection of *B. melitensis* was 18.5% (51) and *B. suis* was 34.1% (94). However, dogs recorded a 31.3% (139) prevalence of *B. abortus* and a 7.9% (35) prevalence of *B. canis* (Figure 3).

There was diversity in the occurrence of *Brucella* biotype in the six regions of Nigeria. The majority of the *brucellae* detected across the six zones were un-typed. In the NC, un-typed *Brucella* spp. was 1.4% (1234), while the NE was 87.4% (887), the NW was 89% (3460), the SE was 43.2% (125), and the SW was 53.1% (621). The NC region also reported other *Brucella* spp., which included *B. abortus* 18.7% (243), *B. melitensis* 48.7% (175), and *B. suis* 41.5% (39). The NE region reported *B. abortus* 12.4% (110), and *B. melitensis* 2.5% (22). In the NW, *B. abortus* was 8% (314), and *B. melitensis* was 2.8% (111). In Southern Nigeria, SE recorded *B. abortus* at 29.6% (37) and *B. canis at* 27.2% (34). In SS, *B. abortus* represented 29.8% (45), while rates of *B. melitensis* of 33.6% (51) and of *B. suis* of 36.4% (55) were observed. The SW region recorded 46.8% (547) *B. abortus* and 0.1% *B. canis* from 1199 positive samples (Figure 4).

### 3.7. The Trend of Reported Human and Animal Brucella Infection in Nigeria (2001 to 2021)

As seen in this review, the trend of brucellosis in Nigeria revealed the earliest report of human *Brucella* infection within the year under review was 2001 in NE Nigeria [45] (Table 1 and Figure 5). There was a general decline in human brucellosis from 2001 to 2012. However, some level of fluctuation was observed in the trend of the disease between 2013 and 2020, with a decline in 2021. The peak of human *Brucella* infection in Nigeria was observed in 2016 from SE Nigeria [92] (Table 1 and Figure 5). The earliest report of brucellosis in animals was in 2003 in NW Nigeria. Generally, there were fluctuations in the trend of the disease. However, there was a remarkable increase in the trend of the disease in animals from 2008 to 2019 and a sharp decline by 2020. The peak of the disease in animals was observed in 2012.

## 4. Discussion

The economic burden in the livestock industry arising from brucellosis and its morbidity in humans has made this zoonotic disease a global public health challenge [111]. The main objective of this study was to systematically review the literature reporting brucellosis and perform a meta-analysis to estimate the national prevalence of brucellosis in Nigeria. To the best of our knowledge, this is the first exhaustive systemic review on human and animal brucellosis in Nigeria. Of the 99 publications accessed, 14 publications reported *Brucella* infection in humans from 11 states of the 36 states in Nigeria, including the federal capital territory. The remaining 25 states either do not have reports because of the lack of interest in brucellosis, or the reported cases do not meet the inclusion criteria of this review. Most of the studies on human brucellosis were from high-risk occupational groups, especially abattoir workers. The national prevalence of human brucellosis in Nigeria for all methods revealed in this review was 17.6%. However, based on the Rose Bengal plate test (RBPT), as reported by about 80% of the publications accessed for human brucellosis, the seroprevalence of the disease in Nigeria was 15.7% (493/3144). This is similar to the 15.8% recorded in Cameroon [112], but lower when compared to the sub-national study in other sub-Saharan African countries, such as the 44% prevalence recorded in Kenya [113], 31.5% prevalence in Ethiopia [114] and 17% in Uganda [115]. Similarly, the prevalence of human brucellosis is also higher in northern African countries such as Egypt, with a prevalence of 23.9% [116]. The prevalence is higher in Nigeria compared to the prevalence of 1.41% reported in Tanzania [117] and the 3.0% national prevalence in Kenya [118]. The high prevalence of human brucellosis from this review has demonstrated the fact that brucellosis is endemic with a high burden in Nigeria and requires the attention of policymakers and stakeholders in the health sector. The high burden of human brucellosis in Nigeria, as seen in this study, can be attributed to several factors, which include nomadic pastoralists who run open grazing, abattoir workers, especially those slaughtering animals who are in constant contact with animal blood without personal protective equipment (PPE) (Figure 6), and the love of Nigerians for the consumption of animals’ intestinal parts, which are the most likely source of zoonotic transmission of the disease in Nigeria. Unfortunately, the animals that are ready for slaughter in Nigeria’s abattoirs are sometimes not screened for brucellosis. The spectrum of clinical presentation of human brucellosis, which mimics several other febrile illnesses such as rheumatic fever, typhoid, and malaria, has resulted in several misdiagnoses of this disease since malaria and typhoid fever are also endemic in Nigeria. Additionally, brucellosis is not routinely screened in private and public health care facilities, which obscures the detection and true prevalence of the disease.

Although the prevalence of the brucellosis reported in this review was based on serological investigation, no publication reported human brucellosis based on culture and molecular methods. Nevertheless, the overall prevalence of brucellosis was higher at **15.6%** (7178/**46,022**) in Northern Nigeria than in the Southern part at 8.7% (1902/21,740). The differences in the prevalence rate may be due to the number of research articles reporting brucellosis in Northern Nigeria being higher when compared to the reportage in the Southern part of Nigeria. Moreover, easy access to and frequency of contact of abattoir workers with animals, e.g., cattle, donkeys, horses, goats, and sheep and animal products, as well as consumption of raw and untreated fermented animal products, especially milk *(nono),* a habit that is very common in northern Nigeria, contribute to the prevalence rates. Furthermore, the uncontrolled movement of nomadic Fulani (herdsmen) from northern Nigeria has contributed significantly to the spread of the disease to other parts of the country, a fact that has long been documented by other authors [119,120].

The burden of brucellosis in animals, as observed in this review, was highest in pigs, at 28.3%, compared to in other animals such as sheep, at 23.3% and camels, at 20.9%. The lowest prevalence was among chickens, with an 8.4% prevalence recorded. The higher prevalence of the disease in pigs could be attributed to multiple *Brucella* spp. detected from the samples investigated in the SS as reported by Bello-Onaghise et al. [95] based on the culture method. From 30 blood samples, 81 *Brucella* isolates were documented (24 *B. abortus*, 27 *B. melitensis*, and 30 *B. suis*), as well as 68 isolates (19 *B. abortus*, 24 *B. melitensis*, and 25 *B. suis)* from 25 vaginal swabs, giving an overall prevalence of 100% (55/55) as reported in that study. Although the report of co-infection among different *Brucella* species is very rare in literature, only Bello-Onaghise et al. [95] have been able to document this in Nigeria to the best of our knowledge. In this review, the occurrence of brucellosis in animals varied from region to region and animal to animal. Interestingly, despite the higher occurrence of the disease in Northern Nigeria, the SS geopolitical zone of Nigeria reported the highest prevalence of brucellosis at 48.6% (206/424) in animals from four publications. The lowest occurrence was observed in the SE, with a prevalence of 3.9%, while the SW was 7%. Other studies included the NC at **16.6%,** the NE at 18%, and the NW at 14.2%. This study also showed evidence of geographical variations in the prevalence of brucellosis not only in the sources of samples but also in the types of samples evaluated. Investigation of brucellosis from different samples across the country revealed the prevalence of the disease estimated to range from 5% in the aborted foetus to 50% in hygroma fluid. Bloodstream infection with brucellosis was estimated at 13.5%, and its prevalence in milk was estimated at 10.2%. The prevalence of brucellosis from vaginal swabs obtained from different animals was 12.4%, while a prevalence of 29.4% was observed in the lymph node of cattle from NW Nigeria. Thus far, only 7 out of the 104 publications reported culture detection methods [15,21,33,36,42,95,97], while a single publication reported molecular methods [78]. The Rose Bengal rapid test is the most popular diagnostic method reported by 11/14 publications on the detection of human brucellosis, while 95/107 publications reported RBPT as the detection method in animal samples. Several publications also reported other diagnostic methods across the country. Although in this review, it was observed that several publications reported funding from donors, grants, and/or direct receipt of diagnostic kits from some laboratories for their studies, unfortunately, we were unable to determine if the donors or the providers of those kits have any influence on the outcomes of their studies.

Four different *Brucella* species were recorded from the systemic review of the literature, including *B. abortus*, *B. melitensis*, *B. suis*, and *B. canis.* However, *B. abortus* biovar 1 has been documented as the most frequently encountered brucellosis in cattle globally [121,122] , while biovar 3 has been reported to be a common biovar in Côte d’Ivoire [123] and Nigeria [33]. *B. melitensis* was mostly reported in humans, and *B. abortus* with biovars 1 and 3 was frequently detected in animals from Nigeria [33,36]. *B. melitensis* was mostly encountered in sheep, goats, and pigs, with few reports in cattle. This agreed with a study conducted in China by Liu et al. [124] in which *B. melitensis* infection was common in sheep and goats. Meanwhile, *B. suis* was observed only in sheep, goats, and pigs. This study also revealed the detection of *B. canis* only in dogs in Nigeria. In Nigeria, a similar distribution pattern of the biodiversity of brucellosis in the different regions occurred. From the Northern to the Southern region, *B. abortus* is the most encountered species in animals, followed by *B. melitensis.* Only the NC and the SS reported *B. suis*, while 99% of the *B canis* detected in dogs were observed in the SE, with only 1% from SW Nigeria. Brucellosis in Nigeria, as seen in this study, is spread across the country and has been responsible for reproductive disease in animals associated with abortion, stillbirth, death of young animals, placenta previa, the birth of immature calves, delayed calving, male infertility, and heavy reductions in milk output.

Most phenotypic detection methods of *Brucella* consist of bacteriological isolation and biochemical identification, which solely relies on a combination of some investigating parameters to characterize suspicious colonies [125]. Unfortunately, this typing method often fails to correctly differentiate some strains, which was seen in the case of the strains reported in Plateau State by Bertu et al. [33]. Thus, there is a need for other combined identification methods, such as the use of molecular tools to characterize *Brucella* species with good discriminatory capacity and the potential to evaluate relationships between species, such as the multiple-locus variable of tandem repeat analysis (MLVA) [126,127]. Future studies on brucellosis should also focus more on the proper use and interpretation of diagnostic testing for animals as recommended by the WHO, FAO, and OIE [1,128,129]. There was stability in the trend of human brucellosis from 2001 to 2012. The trend began to fluctuate from 2013 to 2020, with a decline in 2021. The observed decline in human *Brucella* infection may be connected to the improved abattoir environment in some states in Nigeria when compared to previous years. The peak of human *Brucella* infection in Nigeria was observed in 2016 from SE Nigeria [91] (Table 1 and Figure 5). There was a fluctuation in the trend of the disease in animals from 2003 to 2021, with a remarkable increase in the trend of the disease from 2008 to 2019 and a sharp decline by 2020. The peak of the disease in animals was observed in 2012, though there was no report on the outbreak of the disease in Nigeria in 2012. Furthermore, the number of articles reporting *Brucella* infection in 2012 was also lower than in some of the years under review, while the number of animals screened for brucellosis in 2012 was also lower than the number of samples screened for some years under review. Hence, the increase cannot be attributed to sampling size. Despite the problem of antibiotic resistance being on the rise in brucellae [8], unfortunately, there was no single report on antimicrobial susceptibility testing on the *Brucella* species isolated from both humans and animals, thus making it difficult to determine the pattern of resistance of *Brucella* spp. in Nigeria. The use of quality molecular tools in detecting *Brucella* infection in Nigeria is also lacking and has contributed to many reported un-typed *Brucella* species. There is a need to focus more on culture and molecular methods to determine the epidemiological link between *Brucella* species and its biodiversity from humans and animals, to know the prevailing biovars, and for prompt interventions. The findings from this systemic review would serve as baseline information on the national prevalence of *Brucella* infection in Nigeria. The generated data would help estimate the true burden of *Brucella* infections in Nigeria. As it is today, national data on human and animal *Brucella* infection are lacking in both State and Federal Ministries.

## 5. Limitations

Our inability to determine the prevalence of the disease based on certain variables such as age, sex, cattle herds, breeds of animals, and mortality rate both in humans and animals are part of the limitations of this review. The use of serological methods, as seen in this review, is of national importance because of their usefulness in detecting antibodies against *Brucella* species. However, the results are only reliable when the diagnostic methods and procedures are properly conducted and the outcomes are well interpreted. Interestingly, serological methods have several shortcomings, including the antiseraÁs ability to cross-react with other bacterial pathogens such as *Escherichia coli* 0157, *Salmonella* Urbana group N, *Francisella tularensis*, and *Yersinia enterocolitica* 0:9 [130].

## 6. Conclusions

This systemic review revealed a national prevalence of 19.2% of human *Brucella* infection and 13.1% prevalence of *Brucella* infection in common domestic animals, respectively, from 2001 to 2021. This study also indicated that only 27 states, including the FCT, reported brucellosis in Nigeria across the six geopolitical zones. Our study also revealed the highest occurrence of brucellosis is in Northern Nigeria. Four *Brucella* species, *B. abortus*, *B. melitensis*, *B. suis*, and *B. canis*, have been reported in Nigeria. Culture and molecular methods of detection of brucellosis and reports of antimicrobial susceptibility testing remain a conjecture. This review will help researchers redirect their research focus and serve as a guide for policymakers on measures for managing brucellosis in Nigeria and other sub-Saharan countries.

The need for improved sanitary conditions of the abattoirs, the use of personal protective equipment by animal handlers, vaccination of animals against bovine brucellosis, and ranching of animals to curb the spread of the disease should be paramount to all the stakeholders.

## Figures and Tables

**Figure 1 vetsci-09-00384-f001:**
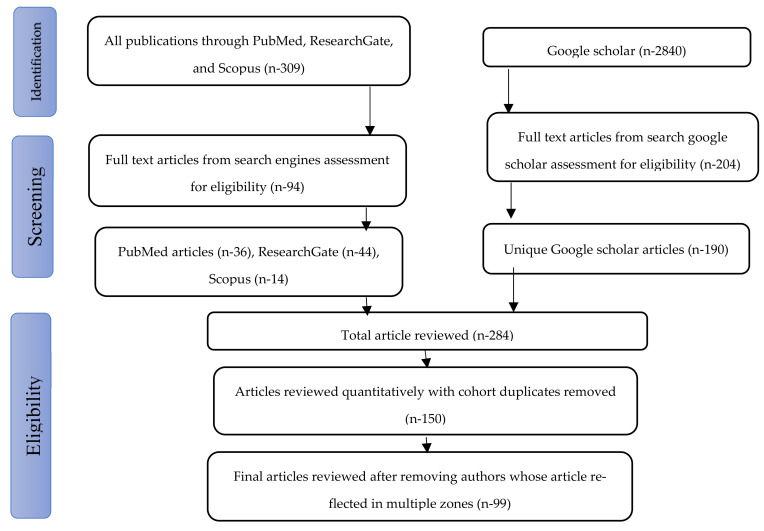
Flowchart of systemic review of human and animal brucellosis for selection of eligible articles.

**Figure 2 vetsci-09-00384-f002:**
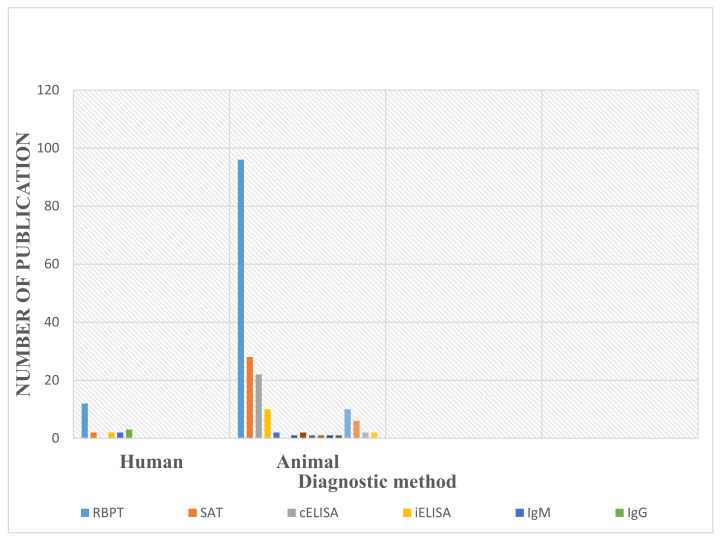
Diagnostic methods used in the detection of human and animal brucellosis in Nigeria.

**Figure 3 vetsci-09-00384-f003:**
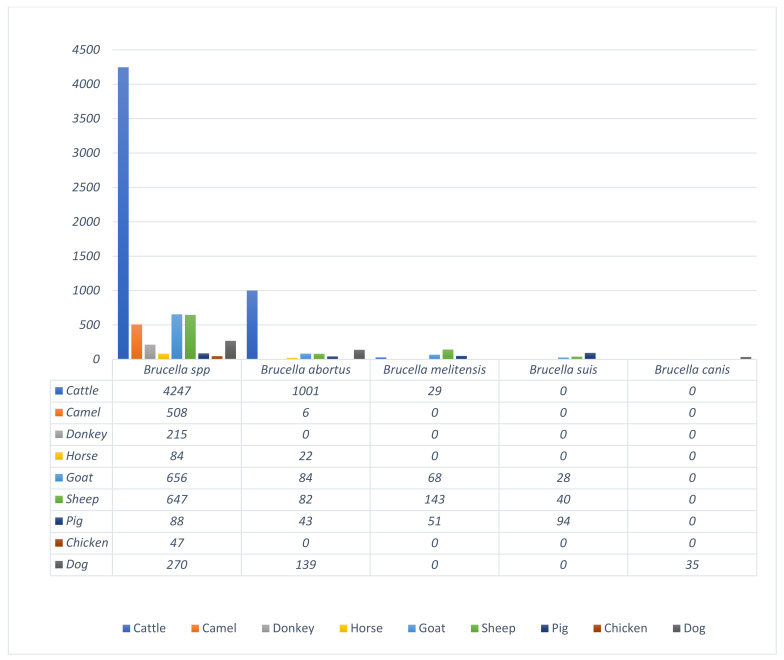
Reported *Brucella* serotypes from different domestic animals studied in Nigeria (2001–2021).

**Figure 4 vetsci-09-00384-f004:**
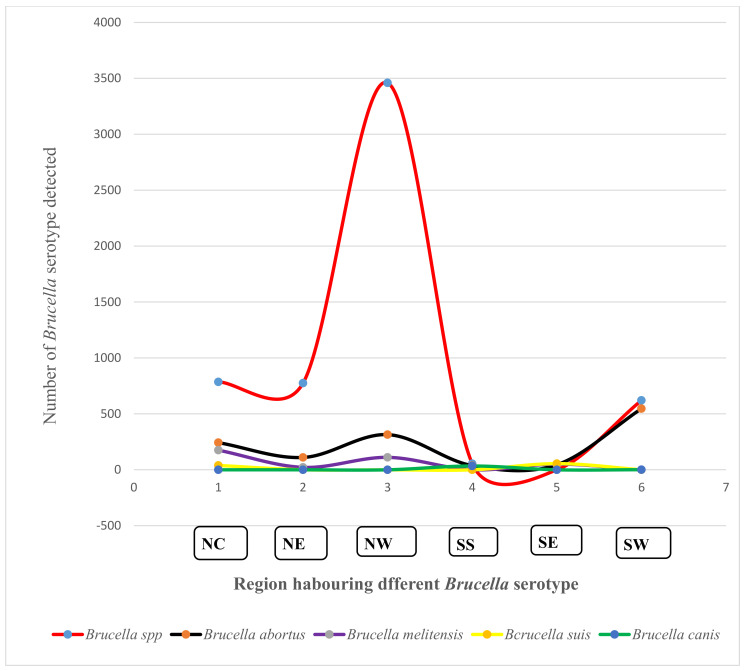
Distribution of prevailing *Brucella* serotypes from domestic animals reported from the six regions of Nigeria (2001–2021).

**Figure 5 vetsci-09-00384-f005:**
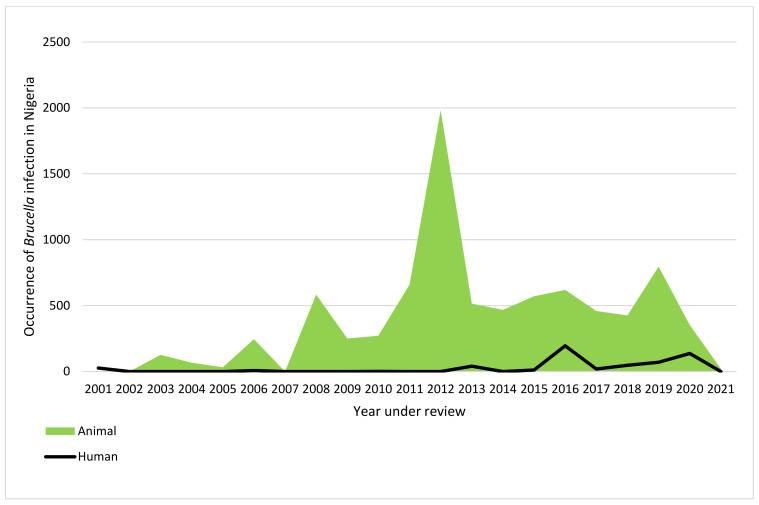
Trend of human and animal brucellosis in Nigeria from 2001 to 2021.

**Figure 6 vetsci-09-00384-f006:**
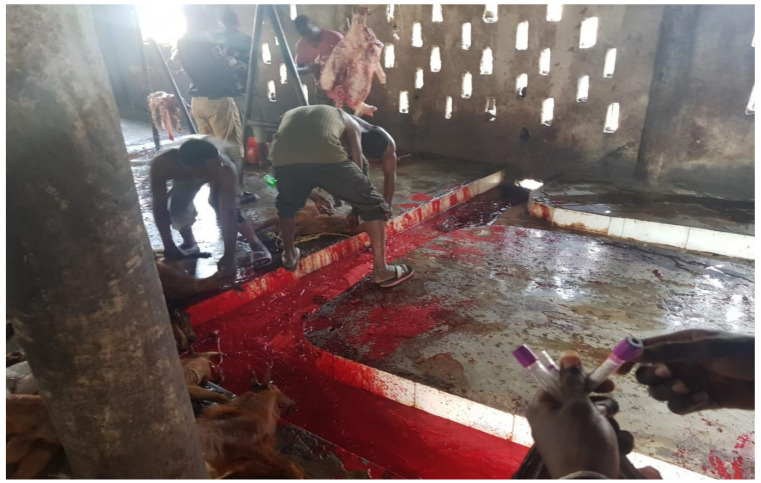
Typical abattoir in Nigeria showing butcher slaughtering animals without personal protective equipment.

**Table 1 vetsci-09-00384-t001:** Seroprevalence of reported human brucellosis in Nigeria (2001–2021).

Region	State	No of Samples Tested	No of Positive Sample	Seroprevalence %	Detection Methods							*Brucella* Type			Reference
					RBPT	SAT	cELISA	iELISA	IgM	IgG	IgG/IgM ELISA	*Brucella* spp.	*Brucella abortus*	*Brucella* *melitensis*	
NC	Kwara	189	42	22.2%	42	0	0	0	0	0	0	42	0	0	[19]
	Abuja	224	40	17.9%	40	0	0	0	22	18	0	40	0	0	[30]
	Nasarawa	160	16	10%	16	0	0	0	0	0	0	16	0	0	[31]
**Total**		**573**	**98**	**17.1%**	**98**	**0**	**0**	**0**	**22**	**18**	**0**	**98**	**0**	**0**	
NE	Bauchi	285	95	33.3%	95	0	0	0	6	18	0	95	0	0	[57]
	Not indicated	500	26	5.2%	26	0	0	0	0	0	0	26	0	0	[45]
	Borno	106	4	3.8%	4	0	0	0	0	0	0	4	0	0	[50]
	Borno	100	11	11%	11	0	0	0	0	0	0	0	11	0	[56]
**Total**		**990**	**136**	**13.7%**	**136**	**0**	**0**	**0**	**6**	**18**	**0**	**125**	**11**	**0**	
NW	Kaduna	1	1	100%	0	1	0	0	0	0	0	0	1	0	[66]
	Kaduna	100	19	19%	0	0	0	19	0	0	0	19	0	0	[72]
	Sokoto	137	1	0.7%	1	0	0	0	0	0	0	0	1	0	[76]
**Total**		**238**	**21**	**8.8%**	**1**	**1**	**0**	**19**	**0**	**0**	**0**	**19**	**2**	**0**	
SE	Enugu	682	195	28.6%	195	0	0	0	0	0	0	195	0	0	[91]
	Total	682	195	28.6%	195	0	0	0	0	0	0	195	0	0	
SS	Akwa Ibom	228	70	30.7%	29	0	0	0	0	41	0	0	0	70	[96]
**Total**		**228**	**70**	**30.7%**	**29**	**0**	**0**	**0**	**0**	**41**	**0**	**0**	**0**	**70**	
SW	Lagos	422	27	6.4%	27	0	0	3	0	0	0	27	0	0	[105]
	Oyo	11	7	63.6%	7	0	0	0	0	0	0	7	0	0	[110]
**Total**		**433**	**34**	**7.9%**	**34**	**0**	**0**	**3**	**0**	**0**	**0**	**34**	**0**	**0**	
**Grand Total**		**3144**	**554**	**17.6%**	**493**	**1**	**0**	**22**	**28**	**77**	**0**	**471**	**13**	**70**	

NC: North-Central, NE: North-East, NW: North-West, SE: South-East, SS: South-South, SW: South-West, RBPT: Rose Bengal plate test, SAT: serum agglutination test, cELISA: competitive enzyme-linked immunosorbent assay, iELISA: indirect enzyme-linked immunosorbent assay, IgM: immunoglobulin M, IgG: immunoglobulin.

**Table 2 vetsci-09-00384-t002:** National prevalence of brucellosis for different domestic animals in Nigeria (2001–2021).

Animal	Variable	The Six Geopolitical Zones in Nigeria						Total Nationwide	National Prevalence %	Odd Ratio	95% CL	Z Statistics	*p*-Value	Number of Publications
		NC	NE	NW	SE	SS	SW							
Cattle	Sample screened	**3213**	5253	19,648	1567	369	12,458	**42,508**	**12.2%**	**0.1390**	**0.1346** **to 0.1435**	**120.853**	<0.0001	43
	+ ve samples	277	1154	2617	51	151	937	5187						
	Regional prevalence	**8.6%**	**22%**	**13.3%**	**3.3%**	**40.9%**	**7.5%**							
Donkey	Sample screened	0	2101	0	0	0	0	2101	10.2%	0.2427	0.2046 to 0.2879	16.247	<0.0001	4
	+ ve samples	0	215	0	0	0	0	215						
	Regional prevalence	**0%**	**10.2%**	**0%**	**0%**	**0%**	**0%**							
Camel	Sample screened	0	1267	1192	0	0	0	2459	20.9%	0.2024	0.1796 to 0.2281	26.196	<0.0001	5
	+ ve samples	0	398	116	0	0	0	514						
	Regional prevalence	**0%**	**31.4%**	**9.7%**	**0%**	**0%**	**0%**							
Horse	Sample screened	86	100	400	402	0	0	988	10.7%	0.0941	0.0729 to 0.1214	18.190	<0.0001	8
	+ ve samples	24	16	54	12	0	0	106						
	Regional prevalence	**27.9%**	**16%**	**13.5%**	**3%**	**0%**	**0%**							
Goat	Sample screened	2134	831	2191	340	0	2813	8309	10.2%	0.1013	0.0931 to 0.1102	53.357	<0.0001	20
	+ ve samples	410	75	303	12	0	46	846						
	Regional prevalence	**19.2%**	**9%**	**13.8%**	**3.5%**	**0%**	**1.6%**							
Sheep	Sample screened	1308	57	2449	0	0	94	3908	23.3%	0.2831	0.2569 to 0.3119	25.465	<0.0001	22
	+ ve samples	322	14	562	0	0	14	912						
	Regional prevalence	**24.6%**	**24.6%**	**22.9%**	**0%**	**0%**	**14.9%**							
Pig	Sample screened	369	0	0	351	55	200	975	18.7%	0.1731	0.1420 to 0.2110	17.371	<0.0001	5
	+ ve samples	125	0	0	2	55	0	182						
	Regional prevalence	**34%**	**0%**	**0%**	**0.6%**	**100%**	**0%**							
Chicken	Sample screened	275	730	0	410	0	140	1555	8.4%	0.0298	0.0213 to 0.0417	20.498	<0.0001	3
	+ ve samples	10	18	0	14	0	5	47						
	Regional prevalence	**3.6%**	**2.5%**	**0%**	**3.4%**	**0%**	**3.6%**							
Dog	Sample screened	492	374	200	123	0	1102	2291	19.4%	0.2337	0.2037 to 0.2680	20.767	<0.0001	8
	+ ve samples	115	76	43	34	0	176	444						
	Regional prevalence	**23.4%**	**20.3%**	**21.5%**	**27.6%**	**0%**	**16%**							

NC: North-Central, NE: North-East, NW: North-West, SE: South-East, SS: South-South, SW: South-West.

**Table 3 vetsci-09-00384-t003:** Regional distribution of reported cases of *Brucella* infection in different samples obtained from common domestic animals in Nigeria (2001–2021).

Region	Types of Animals	Types of Samples Investigated											
		Blood		Milk		Vaginal Swab		Hygroma Fluid		Aborted Foetus		Lymph Node	
		No Tested	+ve Sample	Not Tested	+ve Sample	No Tested	+ve Sample	No Tested	+ve Sample	No Tested	+ve Sample	No Tested	+ve Sample
NC	Cattle	**2397**	212	428	52	374	11	4	1	10	1	0	0
	Donkey	0	0	0	0	0	0	0	0	0	0	0	0
	Camel	0	0	0	0	0	0	0	0	0	0	0	0
	Horse	77	20	0	0	8	3	1	1	0	0	0	0
	Goat	2099	410	18	0	17	0	0	0	0	0	0	0
	Sheep	1242	322	20	0	44	0	0	0	2	0	0	0
	Pig	366	125	0	0	0	0	0	0	3	0	0	0
	Chicken *	275	10	0	0	0	0	0	0	0	0	0	0
	Dog	483	115	4	0	0	0	0	0	5	0	0	0
**Total**		**6939**	**1214**	**470**	**52**	**443**	**14**	**5**	**2**	**20**	**1**	**0**	**0**
NE	Cattle	5047	1120	144	27	56	3	6	4	0	0	0	0
	Donkey	2101	215	0	0	0	0	0	0	0	0	0	0
	Camel	1267	398	0	0	0	0	0	0	0	0	0	0
	Horse	100	16	0	0	0	0	0	0	0	0	0	0
	Goat	831	75	0	0	0	0	0	0	0	0	0	0
	Sheep	28	4	8	7	21	3	0	0	0	0	0	0
	Pig	0	0	0	0	0	0	0	0	0	0	0	0
	Chicken *	730	18	0	0	0	0	0	0	0	0	0	0
	Dog	374	76	0	0	0	0	0	0	0	0	0	0
**Total**		**10,478**	**1922**	**152**	**34**	**77**	**6**	**6**	**4**	**0**	**0**	**0**	**0**
NW	Cattle	16,008	2456	3307	111	161	0	2	0	0	0	170	50
	Donkey	0	0	0	0	0	0	0	0	0	0	0	0
	Camel	1192	116	0	0	0	0	0	0	0	0	0	0
	Horse	400	54	0	0	0	0	0	0	0	0	0	0
	Goat	1937	214	254	89	0	0	0		0	0	0	0
	Sheep	2248	522	201	40	0	0	0	0	0	0	0	0
	Pig	0	0	0	0	0	0	0	0	0	0	0	0
	Chicken *	0	0	0	0	0	0	0	0	0	0	0	0
	Dog	200	43	0	0	0	0	0	0	0	0	0	0
**Total**		**21,985**	**3405**	**3762**	**240**	**161**	**0**	**2**	**0**	**0**	**0**	**170**	**50**
SE	Cattle	1566	51	0	0	0	0	1	0	0	0	0	0
	Donkey	0	0	0	0	0	0	0	0	0	0	0	0
	Camel	0	0	0	0	0	0	0	0	0	0	0	0
	Horse	402	12	0	0	0	0	0	0	0	0	0	0
	Goat	340	12	0	0	0	0	0	0	0	0	0	0
	Sheep	0	0	0	0	0	0	0	0	0	0	0	0
	Pig	351	2	0	0	0	0	0	0	0	0	0	0
	Chicken *	410	14	0	0	0	0	0	0	0	0	0	0
	Dog	123	34	0	0	0	0	0	0	0	0	0	0
**Total**		**3192**	**125**	**0**	**0**	**0**	**0**	**1**	**0**	**0**	**0**	**0**	**0**
SS	Cattle	354	149	0	0	14	1	1	1	0	0	0	0
	Donkey	0	0	0	0	0	0	0	0	0	0	0	0
	Camel	0	0	0	0	0	0	0	0	0	0	0	0
	Horse	0	0	0	0	0	0	0	0	0	0	0	0
	Goat	0	0	0	0	0	0	0	0	0	0	0	0
	Sheep	0	0	0	0	0	0	0	0	0	0	0	0
	Pig	30	81	0	0	25	68	0	0	0	0	0	0
	Chicken *	0	0	0	0	0	0	0	0	0	0	0	0
	Dog	0	0	0	0	0	0	0	0	0	0	0	0
**Total**		**384**	**230**	**0**	**0**	**39**	**69**	**1**	**1**	**0**	**0**	**0**	**0**
SW	Cattle	11,234	689	1224	248	0	0	0	0	0	0	0	0
	Donkey	0	0	0	0	0	0	0	0	0	0	0	0
	Camel	0	0	0	0	0	0	0	0	0	0	0	0
	Horse	0	0	0	0	0	0	0	0	0	0	0	0
	Goat	2813	46	0	0	0	0	0	0	0	0	0	0
	Sheep	94	14	0	0	0	0	0	0	0	0	0	0
	Pig	200	0	0	0	0	0	0	0	0	0	0	0
	Chicken *	140	5	0	0	0	0	0	0	0	0	0	0
	Dog	1102	176	0	0	0	0	0	0	0	0	0	0
**Total**		**15,583**	**930**	**1224**	**248**	**0**	**0**	**0**	**0**	**0**	**0**	**0**	**0**
**Grand Total**		**58,561**	**7826**	**5608**	**574**	**720**	**89**	**14**	**7**	**20**	**1**	**170**	**50**

***** Chicken (Avian, Duck, Turkey, Guinea fowl), NC: North-Central, NE: North-East, NW: North-West, SE: South-East, SS: South-South, SW: South-West.

## Data Availability

Not applicable.

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
