# Peer review of "Human and Animal Brucellosis in Nigeria: A Systemic Review and Meta-Analysis in the Last Twenty-One Years (2001–2021)"

_vetsci, 2022, doi:10.3390/vetsci9080384_

Round 1

Reviewer 1 Report

Brucellosis is an important zoonotic infections having enormous economic impact on livestock sector and human health sector. The endemicity of the disease in Nigeria requires an up to date information to develop preventive and control strategies. Therefore, the current review article provides valuable information. There are certain minor issues that need to be addressed.

1. Table 2 and 3: the number of cattle tested in “NC” regions is 2930. However total number of  tested cattle in NC region in table 3 does not add up to 2930, but they add up to 2554. So, the authors should once again check all data for accuracy.

2. Figure 5: the marker shape is same for both humans and animals same. Change human or animal marker shape to square or diamond.

3. Line 305: The sentence starting with “while……..” should be connected with first sentence.

Author Response

RESPONSE TO REVIEWER 1

Query: Table 2 and 3: the number of cattle tested in “NC” regions is 2930. However total number of tested cattle in NC region in table 3 does not add up to 2930, but they add up to 2554. So, the authors should once again check all data for accuracy.

Response: The number of cattle tested as well as the number of positive samples in the NC have been corrected. The odd ratio, Z- statistics as well as the p. value have been corrected. Which are highlighted in  green color

Reviewer 2 Report

Dear Authors

Greetings

Please read the attached doc with some suggestions using Adobe.

Regards

Author Response

RESPONSE TO REVIEWER 2

The corrections are highlighted in green color in the manuscript where applicable

  1. Did you apply some prevalence study quality appraisal?

Yes. The critical appraisal tool for prevalence studies developed and tested by Munn et al. 2014 was used

  1. Did you apply the MetaXL version 5.3. add-in for meta-analysis in Microsoft Excel?

MetaEasy was used as an add-in in Microsoft Excel

  1. Did you perform some software for meta/analysis?

All the necessary steps required in conducting a meta-analysis were followed

  1. Suggestion to find other graphic models for figures

The Figures have been modified generally for heterogenicity of the figures appropriately where applicable.

  1. Standardization of bibliographic

     The referencing style of veterinary science has been infected, and all necessary corrections have been made
